# Microstructure Evolution and Its Correlation with Performance in Nitrogen-Containing Porous Carbon Prepared by Polypyrrole Carbonization: Insights from Hybrid Calculations

**DOI:** 10.3390/ma15103705

**Published:** 2022-05-22

**Authors:** Shanshan Li, Fang Bian, Xinge Wu, Lele Sun, Hongwei Yang, Xiangying Meng, Gaowu Qin

**Affiliations:** 1College of Sciences, Northeastern University, Shenyang 110819, China; shanshanneu@163.com (S.L.); wxg_work@163.com (X.W.); 2Key Laboratory for Anisotropy and Texture of Materials (MoE), School of Materials Science and Engineering, Northeastern University, Shenyang 110819, China; wenxh@atm.neu.edu.cn (F.B.); qingw@smm.neu.edu.cn (G.Q.); 3College of Information Science and Engineering, Northeastern University, Shenyang 110819, China; cumtsll@163.com; 4State Key Laboratory of Advanced Technologies for Comprehensive Utilization of Platinum Metals, Kunming Precious Metals New Materials Technology Co., Ltd., Kunming 650106, China; yhw@ipm.com.cn; 5The State Key Laboratory of Rolling and Automation, Northeastern University, Shenyang 110819, China

**Keywords:** nitrogen-containing porous carbon, carbonization, polypyrrole, machine learning, reactive molecular dynamics

## Abstract

The preparation of nitrogen-containing porous carbon (NCPC) materials by controlled carbonization is an exciting topic due to their high surface area and good conductivity for use in the fields of electrochemical energy storage and conversion. However, the poor controllability of amorphous porous carbon prepared by carbonization has always been a tough problem due to the unclear carbonation mechanism, which thus makes it hard to reveal the microstructure–performance relationship. To address this, here, we comprehensively employed reactive molecular dynamics (ReaxFF-MD) simulations and first-principles calculations, together with machine learning technologies, to clarify the carbonation process of polypyrrole, including the deprotonation and formation of pore structures with temperature, as well as the relationship between microstructure, conductance, and pore size. This work constructed ring expressions for PPy thermal conversion at the atomic level. It revealed the structural factors that determine the conductivity and pore size of carbonized products. More significantly, physically interpretable machine learning models were determined to quantitatively express structure factors and performance structure–activity relationships. Our study also confirmed that deprotonation preferentially occurred by desorbing the dihydrogen atom on nitrogen atoms during the carbonization of PPy. This theoretical work clearly reproduces the microstructure evolution of polypyrrole on an atomic scale that is hard to do via experimentation, thus paving a new way to the design and development of nitrogen-containing porous carbon materials with controllable morphology and performance.

## 1. Introduction

Since the first report on nitrogen-containing porous carbon (NCPC) in 2005 [1], an enormous amount of research has been performed to investigate new classes of NCPCs with higher conductivity, larger surface areas [2,3], and tunable pore sizes for potential applications in gas adsorption [4,5,6], catalysis [7,8,9,10,11,12,13,14,15,16], and energy storage and conversion devices [17,18,19]. When selecting the precursors for preparing NCPCs, aromatic conductive polypyrrole (PPy) is the preferred material due to its simple synthesis, high conductivity, environmental stability, and biocompatibility [20,21,22,23,24,25].

The typical method for preparing NCPCs is the template method [26,27,28]. The template (SiO_2_, Al_2_O_3_, or zeolite) can be removed by chemical etching by reacting with acids and bases (HCl, NaOH, or HF) to form soluble substances or gases. Therefore, the template method inevitably consumes acids and bases, creates pollution, and results in high costs. Alternatively, the preparation of NCPCs can be simplified by the one-step carbonization of conductive polymers [29,30,31], a process that has high N-doping efficiency and is environmentally friendly and convenient. However, the direct carbonization of conductive polymers produces amorphous NCPCs, whose microstructures and properties are difficult to control.

P. Bairi et al. pointed out that thermal conversion products are related to carbon source and temperature [32,33,34]. The performance of materials is undoubtedly associated with their internal microstructures. Therefore, to improve the controllability of amorphous NCPCs prepared by carbonization, it is necessary to clarify the microstructural characterization of carbonized products, the evolution of the microstructure with temperature and time, and the relationship between the microstructure and its properties.

Much attention has recently been given to designing novel NCPCs with tunable nitrogen contents and understanding the structure–property relationship, which is crucial for enhancing their performance [29,30,35,36,37]. However, current knowledge about the microstructure modulation of conversion products comes almost exclusively from experiments. Real-time information on microstructure evolution during carbonization is still contained in a black box. We previously established a ring structure characterization method for porous carbon materials based on ReaxFF molecular dynamics (ReaxFF-MD) simulations [38]. In the current work, we extended the previous work to provide a solution for regulating porous carbon properties by digging into quantitative relationships between the ring structure and the properties of porous carbon materials.

In the current work, we employed a variety of theoretical methods to comprehensively study the preparation process of NCPCs via the carbonization of polypyrrole, including microstructure characterization, microstructure changes with temperature, and the relationship between the microstructure, conductivity, and pore size. Using reactive molecular dynamics (ReaxFF-MD) simulations, we constructed a ring characterization scheme for the evolution of the microstructure during PPy carbonization. We used machine learning techniques to determine the structural factors describing the carbonized tissue’s conductance and pore size parameters and further established quantitative mathematical models. We revealed the deprotonation sites and pathways during PPy carbonization based on MD simulations and first-principles calculations. This theoretical knowledge is conducive to the optimal design of porous carbon materials for specific applications. By establishing the functional relationship between theory and experiment, we hope to find a way to realize the expected ring structure in nanomaterials to develop porous carbon materials with controllable morphology and excellent performance. 

## 2. Computational Details and Models

In this work, based on the needs of the research content, the following calculation methods were adopted to study PPy carbonization.

### 2.1. Reax-FF MD Simulations

The simulation box contained four polypyrrole chains. Appendix A shows the pyrrole molecule and pyrrole chain. The periodic boundary conditions were used in the molecular dynamics simulations, ensuring the constant number of particles in the simulated system and eliminating boundary effects. The size of the container was adjusted so that the initial density of the PPy system at *T* = 300 K was 1.51 g/cm [3,39,40], which is consistent with the experimental density. The experimental density refers to that used when constructing the initial amorphous model. It is the ratio of mass to volume, and this kind of density has been commonly applied to molecular dynamics simulations. In the heating simulations, we first performed energy minimization with ReaxFF-MD simulations. The energy minimization was terminated when the difference between the energy and the force was less than 10^−6^. Then, the PPy systems were equilibrated at 300 K for 100 ps, and we constructed 6 sample structures from the 300 K trajectory within the last 10 ps of the simulation. Subsequently, the samples were heated to 3000 K with a heating rate of 5 K/ps using the ReaxFF-MD method. In the thermostatic simulations, canonical (NVT) ensemble annealing simulations were performed for 6 ns at constant temperatures. For heating and thermostatic simulations, the generated gas products (H_2_, CH_4_, HCN, NH_3_, etc.) were removed from the system every 1 ns to remain consistent with the experimental procedure. 

ReaxFF-MD simulations were carried out with the Large-scale atomic molecular massively parallel simulator (LAMMPS) package [41]. Simulation snapshots were generated using visual molecular dynamics (VMD) [42] and OVITO [43] software, and the pore size was analyzed using the Zeo++ tool [44]. The ReaxFF_C-2019_ potential [45] was applied to the simulation process. ReaxFF_C-2019_ was developed to characterize the dissociation and formation of chemical bonds; thus, it was believed to be a suitable forcefield for the previous study of polymer precursors [46,47] and the current investigation of PPy carbonization.

### 2.2. DFT-NEB Calculations for Dehydrogenation

All DFT calculations were performed with the Vienna Ab-initio Simulation Package (VASP) [48,49,50]. The projector augmented wave (PAW) method and a van der Waals force-corrected Perdew–Burke–Ernzerhof (PBE) functional (DFT-D3) were applied to describe the interactions between the valence electrons and ionic cores and the electron exchange–correlation energies, respectively [51,52]. The plane wave cut-off energy of 500 eV and 2 × 2 × 1 Gamma-Center k-points sampling were chosen as the optimization and CINEB calculation parameters [53,54,55]. The energy and force convergence criteria were set to 10^−7^/10^−5^ eV and 0.05/0.02 eV/Å, respectively.

### 2.3. NEGF-DFT Calculations for Electronic Conductance 

A two-probe transport model was constructed to calculate the electronic conductance, the schematic diagram of which is illustrated in Appendix A In the electrode region, a slab containing the two layers of Al atoms was used as the electrode’s repetitive unit. The Al slab was extended along the *z*-direction to provide the bias and to collect the current. Different rings acted as the central scattering region. The buffer region consisted of five layers of Al atoms in the *z*-direction to isolate possible interactions between the electrode and the central scattering region. The optimized distance between the rings and the buffer region was 1.2 Å, based on the results of the DFT total energy calculation. Changes in the energy with distance were shown in Appendix A Finally, a vacuum region of no less than 20 Å in the *x*-direction and *y*-direction was added to the transport model to screen the interactions between the periodic images of the model. We successfully applied a similar transport model in our previous study of the electronic conductance of a Ni/C interface [56].

The NANODCAL package [57,58] was used to calculate the rings’ conductance. The software was based on non-equilibrium Green’s function density functional theory (NEGF-DFT). In the current study, the local density approximation (LDA) for the exchange–correlation functional and a double-zeta polarized (DZP) atomic orbital basis set was adopted [59,60]. In the self-consistent calculations, the convergence criterion was set to less than 10^−5^ Hartree for the density matrix of every element.

### 2.4. Machine Learning for Structural Factors and Tissue-Performance Relationships

Compressed, sensing-based, data-driven algorithm SISSO [61] was used to dig the structural factors and tissue-performance relationships in PPy. This algorithm has been applied to find physical descriptors in the materials and chemistry fields [62]. By constructing a high-dimensional feature space and solving the sparse solution of the linear model, the best descriptor for the target quantity can be found efficiently. Each feature is a function based on a physical quantity and has certain interpretability. The final descriptor is a combination of key features. The algorithm steps can be described as follows. First, we took the input feature related to the target attribute as the starting point of feature space Ф*_0_*, and then we recursively performed the algebraic operation to expand the feature space. After several iterations, Ф*_0_* was expanded into a huge Ф*_n_*, and there was an exponential relationship between the size of the feature space Ф*_n_* and the number of iterations *n*. 

Subsequently, the descriptors were filtered from the feature space using deterministic independence screening (SIS) and sparsity operator (SO) methods. SIS is a method for reducing the dimensions of high-dimensional spaces. It calculates the inner product of target properties and features, evaluates their correlation, and selects features with high correlation as subspaces. After dimensionality reduction, the SO method was carried out to determine the optimal dimensional descriptor. The main parameters, set artificially, were the dimension of two descriptors (the number of non-constant items in the Formula) and the complexity of descriptors (the number of operators in descriptors and descriptor complexity). Data sets of the target properties and features required for structural factors and tissue-performance mining were provided in Appendix A

## 3. Results and Discussion

We constructed six samples with different initial structures as research subjects to eliminate individual differences during the simulation. Similar to the structure of porous carbon [38], we found multiple C-C and C-N bonds in the nitrogen-containing carbon formed by the carbonization of PPy, i.e., bonds in five-membered rings (5-M) and six-membered rings (6-M). Thus, we followed the ring structure characterization method and further clarified the labeling principle of the ring structure in NCPCs. We use (C-C)*_n_* and (C-N)*_n_* to represent the different C-C and C-N bonds, where *n* is the coordination number of rings. For example, (C-N)*_5_* is a C-N bond in a 5-M ring. Accordingly, 5-MN_1_ denotes five-membered carbon rings containing one nitrogen atom. In the following sections, this ring notation is used to characterize the microstructure of carbonized PPy.

### 3.1. Tissue Evolution and Critical Temperature during PPy Carbonization

We first adopted heating simulations to roughly observe the tissue evolution during the carbonization process of PPy. A total of 6 samples were equilibrated at 300 K by 100 ps constant-temperature simulations. The temperature and energy changes with time verified the equilibrium states, as shown in Appendix A Then, the samples were heated to the target temperature (3000 K) with a 5 K/ps heating rate. 

Figure 1a shows the variation of H_2_ molecules with temperature during the heating process of the 6 sample structures. We found that no H_2_ was produced in the system when the temperature was below 700 K, indicating that C–H or N–H bonds were not broken. When the temperature was raised to above 700–1800 K, deprotonation occurred, and the deprotonation rate was linear with temperature. At 1800–2800 K, the H_2_ content remained almost unchanged, while the number of H_2_ increased significantly at 2800 K. Actually, deprotonation below 2800 K occurred at the nitrogen sites. Above that temperature, hydrogen on carbon was also desorbed. The deprotonation is closely related to the dehydrogenation mechanism, explained in detail in Section 3.4. In addition, we found no structure evolution below 1500 K, according to the ring structure analysis in Figure 1b. Above 1500 K, the number of 5-M rings decreased, and other ring structures, such as 6-M, 7-M, and 8-M rings, appeared in the systems. In general, through the heating simulation of PPy, we could roughly determine that the theoretical dehydrogenation temperature is 700 K, and the critical temperature of pyrolysis is 1500 K.

Subsequently, a series of detailed thermostatic simulations were performed to verify the heating simulation results. In these studies, the samples were balanced for 6 ns in 200 K intervals from 1300 to 2500 K. Figure 2 shows the evolution of the average number of rings, with the simulation time at each temperature. 

Below the 1500 K, the number of 5-membered rings in the system did not change, and no other stable types of rings were formed. At 1700 K, deprotonation occurred in the early simulation stage, then the number of 5-M rings decreased slowly, and 6-M rings appeared in the later stage of the simulation, indicating the occurrence of the pyrolysis reaction. As the temperature rose, the variety and number of new rings increased. At 1900 K, we observed new rings (6-M, 7-M, and 8-M). The number of new rings, especially 6-M rings, was negatively correlated with the number of 5-M rings. However, we found that 8-M rings were not stable and disappeared at the end of thermostatic simulations at 1900 K.

At 2100 and 2500 K, the rings changed more dramatically and in different ways. The number of 5-M rings decreased rapidly, then rose, and finally remained constant. After structural analysis, we confirmed that this phenomenon originated from the formation of pure carbon 5-M rings after the emission of nitrogen atoms in the later carbonation stage, which inversely increased the total number of 5-M rings in the system. The simulation results showed that the number of 6-M rings significantly suppressed the number of 5-M rings, indicating that graphitic carbon was the dominant form in the system, and the graphitization of PPy was complete. The 8-M rings remained unstable and tended to fade in the later stages of the simulation. The thermostatic simulation results further confirmed that the theoretical critical temperature of the pyrolysis reaction is 1500 K.

In experiments, the deprotonation of PPy materials generally occurs at 450–600 K [63], and the critical temperature of the pyrolysis reaction is usually 1200 K [19]. It should be noted that the theoretical critical temperature is higher than the one monitored in experiments. To follow the tissue evolution in a limited simulation time, we employed an accelerated kinetics method [38,64] in the simulation process, which returned a higher critical temperature. However, we reproduced the tissue evolution observed in the experiment by MD simulations, and accelerated kinetics has been widely accepted and applied in theoretical studies [38,64,65]. As presented below, our theoretical simulations provide essential knowledge and demonstrate the feasibility of the controllable preparation of amorphous NCPCs, which will promote the development of experimental work.

### 3.2. Tissue Correlation with Electronic Conductance

Conductance is a material property with crucial practical value. It shows a significant correlation with tissue during the carbonization of PPy. Still, the understanding of this relationship has not been solved, which poses a substantial obstacle to regulating the conductance through tissue. Now that we have constructed a ring-structure characterization scheme for the carbonized tissue, it is possible to build structure–property relationships between the conductance and the ring structure.

Different from ionic conductance, the calculation of electronic conductance is a rather tricky problem for our systems because ReaxFF-MD provides little information on the transport of electrons. The current model systems are also too large for first-principles conductance calculations based on non-equilibrium Green’s functions. We need a physical quantity to characterize the evolution of conductance with the carbonized tissue to elucidate the structure-performance relationship. 

Since the ring structure has been used as a descriptor of the tissue, we constructed a statistical conductance model based on the proportion of ring structures in the PPy carbonized tissue. The statistical conductance is defined as:(1)$$Con=\sum_{i}w_{i}\sigma_{i}$$
where $w_{i}$ is the weight of different types of rings in the system obtained by MD trajectory analysis, and $\sigma_{i}$ is the conductance of the corresponding rings calculated by the non-equilibrium Green’s function DFT method that is listed in Table 1.

In Figure 3, we plot the variation in the statistical conductance with temperature using the equilibrium structure at each temperature. We know that below the critical temperature, PPy undergoes dehydrogenation. As shown in Figure 3, the statistical conductance decreased gradually with dehydrogenation from 700 to 1500 K because the polypyrrole salt slowly transformed into a less-conductive base [66,67,68]. Then, the pyrolysis reaction occurred, and the structure began to evolve when the simulated temperature was higher than the critical temperature. We found that the statistical conductance increased with PPy pyrolysis from 1500 to 2300 K. The behavior of the statistical conductance below 2300 K is consistent with experiments in which the conductivity of PPy during carbonization showed a quadratic variation with temperature [63,67]. More importantly, our theoretical calculations revealed a phenomenon that was not found in the experimental results due to the temperature limitation. When over-carbonation occurred, the tissue was graphitized, and the conductance instead decreased upon increasing the temperature. Thus, the conductance–temperature curve shows a tissue-dependent optimum conductance during carbonization. 

Although the statistical conductance differed from the real conductivity in the numerical and physical senses, we verified that the statistical model correctly reflected the evolution of conductivity with carbonized tissue by comparing the calculated results with experimental data; thus, it can be further used to explore the tissue correlation with conductivity. 

Because the complexity of the PPy amorphous carbonized structure is beyond the scope of our rational treatment, we introduced a machine learning method to study the dependence of conductance on the structure. In this process, 13 ring structures were treated as machine learning features, and the statistical conductance was the target property. Above the critical temperature, the PPy underwent significant tissue evolution, so we collected feature and target data sets by analyzing and calculating the outputs of molecular dynamics thermostatic simulation at 2100 and 2500 K. 

First, the XGBoost algorithm [69] was used to analyze the weights of the effect of ring structures on conductance. Table 2 shows that the top four features affecting conductance were the 5-M ring, 5-MN_1_ ring, 6-M ring, and 6-MN_1_ ring, accounting for 17.7%, 24.2%, 17.4%, and 9.2% weight, respectively, with a total weight that is close to 70%.

Then, the four features were used to construct the initial feature space Ф_0_. The SISSO approach was executed to mine the functional relationship between the target property and features. In the first iteration of descriptor optimization, 55 candidate features were generated, and 1130 features were generated in the second iteration. The features most closely related to the target property were filtered through sure independence screening (SIS) in each iteration. Then, the formula was fitted by the sparsifying operator (SO) with a specific root mean square error (RMSE). 

Among several structure–activity relationship models produced by machine learning, we chose the formula with the smallest RMSE and the most apparent physical meaning (see Appendix A). The structure–performance relationship between the conductance and microstructure can be presented as:(2)$$Con=\gamma \left(T\right)*\sqrt[{3}]{N_{6-M}/N_{5-MN1}}+B\left(T\right)$$
where γ(T) and B(T) are temperature-dependent parameters that can be obtained by fitting the constant temperature MD simulation data at different temperatures. At 2100 K, γ(T) and B(T) are 0.586 and 0.908, respectively, while they are 0.433 and 1.045 at 2500 K. $N_{6-M}$ is the number of 6-membered carbon rings in the PPy tissue, and $N_{5-MN1}\;$ is the number of 5-membered rings containing 1 nitrogen. Their combination constitutes a structural factor *F* in the carbonized PPy that can be used as a descriptor of conductance. We found that γ(T) and B(T) changed slightly at 2100 K and 2500 K and became approximately 0.5 and 1, respectively. Therefore, an approximate formula of conductance, independent of temperature, may be expressed as:(3)$$Con=0.5*F^{1/3}+1;\;\;\;\;F=N_{6-M}/N_{5-MN1}$$

Formula (3) can represent the relationship between conductance and structural evolution during polypyrrole carbonization at different temperatures. 

Using Formula (2), we plotted the conductance variations with time and the structural factor at 2100 and 2500 K in Figure 4. The data mining results closely reproduced the variation trend of conductance with time at both temperatures. In the physical dimension, γ(T) has the meaning of conductivity density, and B(T) is the conductance of the initial PPy. Therefore, Formula (2) is a physically interpretable machine learning model. The training set used in machine learning for conductance training was listed in Appendix A

The conductance–rings relationship is important for the controlled carbonization of PPy because it can directly obtain nitrogen-containing carbon materials with expected properties through tissue regulation to improve the preparation efficiency and reduce costs. Different kinds of rings can be distinguished in experiments by fitting them to the component peaks in X-ray photoelectron spectroscopy (XPS) spectra [19]. Then, the type and number of rings can be obtained by analyzing the position and intensity of these peaks. Also, carbonized PPys can be characterized by transmission electron microscopy (TEM) [35], powder X-ray diffraction (XRD) [70], Raman spectroscopy [66], and nitrogen adsorption-desorption isotherms [14]. These experimental methods make it feasible to prepare NCPCs via0 the controlled carbonization of PPy.

### 3.3. Tissue Correlation with Pore Size

Pore size is an essential parameter of porous carbon materials, but the evolution mechanism of pore size with ring structure is still unclear. Based on the results presented in Section 3.1, the pore size in the carbonized PPy changes significantly only during the pyrolysis stage. Therefore, we will focus on variations in the pore size distribution (PSD) above the critical temperature (1500 K). 

In Figure 5, we plotted the variation of PSD with temperature using the equilibrium structure at the respective temperature. Below 1500 K, deprotonation led to a gradual increase in the pore size of the system. Pyrolysis-dominated reactions occurred between 1500 and 2300 K, and different types of rings were distributed in the carbonized PPy tissue. Upon increasing the temperature, the degree of carbonization deepened, and the pore size of the structure continued to increase. At 2500 K, PPy was over-carbonized, and we observed apparent graphite-like layered structures. We found that at 2500 K, the pore size first increased, and when the graphite-like structure spread, the pore size gradually decreased. The final reduced pore size at 2500 K was similar to that at a lower temperature (2100 K). More detailed information about microstructure and aperture distribution can be found in Appendix A

By employing a machine learning strategy similar to the one used in the conductance study, we found that the top four features affecting pore size were 5-M ring, 5-MN_1_ ring, 6-M ring, and 6-MN_1_ ring, accounting for 18.7%, 22.4%, 14%, and 13.2% weight, respectively, with a total weight close to 70%, as shown in Table 3. Among several structure–activity relationship models produced by SISSO, we chose the formula with the smallest RMSE and the most apparent physical meaning (see Appendix A). Finally, we dug a pore size–ring relationship for the carbonized PPy as: (4)$$Psd=A\left(T\right)*X+C\left(T\right);\;X=\left(N_{6-m}*N_{6-MN1}-N_{5-MN1}\right)$$

In this model, the pore size is proportional to the structure factor *X*, which combines the number of different ring types, including 6-membered carbon rings $N_{6-m}$, 6-membered rings containing one nitrogen $N_{6-MN1}$ and 5-membered rings containing 1 nitrogen $N_{5-MN1}$. A(T) and *C*(T) are temperature-dependent parameters whose temperature dependence can be fit using MD simulations. Formula (4) is also a physically interpretable machine learning model because A(T) has the physical meaning of pore density, and *C*(T) is the initial pore size. Figure 6 shows the linear change of pore size with the structure factor *X* at 2100 K and 2500 K. At both temperatures, the fitting results reflect the pore evolution within an acceptable error range. The training set for machine learning were provided in Appendix A

### 3.4. Deprotonation Mechanism

The hydrogen content in PPy plays a vital role in conductance and pore size, making it necessary to understand the deprotonation process and mechanism. The deprotonation process was accompanied by hydrogen evolution [66,67,68]. Therefore, we first studied changes in the hydrogen evolution rate and hydrogen content in PPy with temperature. 

As shown in Section 3.1, deprotonation occurred above 700 K. Figure 7a shows the variation of the amount of H_2_ generated over time from 700 to 1700 K. The hydrogen generation rate gradually accelerated upon increasing the temperature. Accordingly, the hydrogen content in PPy decreased as the temperature rose, as shown in Figure 7b, indicating that a higher temperature promoted the deprotonation reaction. 

Further, we studied the deprotonation process and mechanism in detail. Figure 8a–f shows the simulated equilibrium structures of PPy at 700–1700 K. By analyzing the deprotonation pathway, we found that, in the temperature range of 700–1500 K, the hydrogen atom on the nitrogen atom dissociated first to form H_2_. At 1700 K, with the occurrence of the pyrolysis reaction, the hydrogen atoms attached to carbon atoms were transferred to nitrogen atoms and then further dissociated to produce hydrogen. This shows that nitrogen was the deprotonation site. 

At 2100 and 2500 K, we observed the same phenomenon as at 1700 K, in which hydrogen atoms at different initial positions first moved to nitrogen atoms and then generated H_2_. This may be because the N–H bonds broke more preferentially than the C–H bonds, and the migration barrier of the hydrogen atom in tissue was low. Nudged elastic band (NEB) calculations based on density functional theory were performed to explain the hydrogen evolution behavior in the carbonized PPy. The hydrogen evolution path was shown in Figure 9.

Using the saturated tricyclic PPy models, we calculated the dissociation energy of hydrogen at the carbon and nitrogen sites, respectively. We found that the dissociation energy of a single hydrogen atom from carbon was ~1.8 eV higher than that from nitrogen. However, the dissociation energy of dihydrogen atoms was much lower than that of single hydrogen atoms, about 2.0 eV lower, both at the carbon and nitrogen sites. The migration activation energy of the hydrogen atom moving on the rings was ~0.46 eV, which is a lower barrier than the ambient energy (1.5–2.5 eV). Compared with the thermal energy provided by the environment, the current NEB calculations support the notion that deprotonation preferentially occurs by desorbing dihydrogen atoms at nitrogen sites. Appendix A in S.I. provide detailed NEB calculations and discussions.

In Section 3.1, we found that the dehydrogenation rate increased significantly at 2800 K. The hydrogen evolution path at this temperature was shown in Figure 10. By analyzing the production process of H_2_ in the system, we found that the hydrogen at the carbon and nitrogen sites dissociated directly at this temperature. Therefore, the additional deprotonation sites caused by the ambient temperature increased significantly, promoting the hydrogen evolution process. 

## 4. Conclusions

Amorphous porous carbon prepared by carbonization has poor controllability, mainly because of the unclear characterization of carbonized tissue and the microstructure–performance relationship. This manuscript discovered the deprotonation, pyrolysis, and graphitization mechanisms of PPy undergoing carbonization using a variety of theoretical calculations. This computational work constructs a ring expression for PPy thermal conversion at the atomic level. It reveals performance-related structural factors in the carbonized tissue that can quantitatively describe the performance evolution. In addition, our study confirmed that deprotonation preferentially occurred by desorbing dihydrogen atoms at nitrogen atoms, clarifying the deprotonation sites and hydrogen evolution paths during the carbonization of PPy. Our comprehensive atomic-scale study uncovers the feasibility of modulating the microstructure and properties of NCPC materials and the thermal conversion pathways correlated with tissue monitoring and temperature. It also contributes to the design and development of high-surface-area materials with controllable morphology and performance.

## Figures and Tables

**Figure 1 materials-15-03705-f001:**
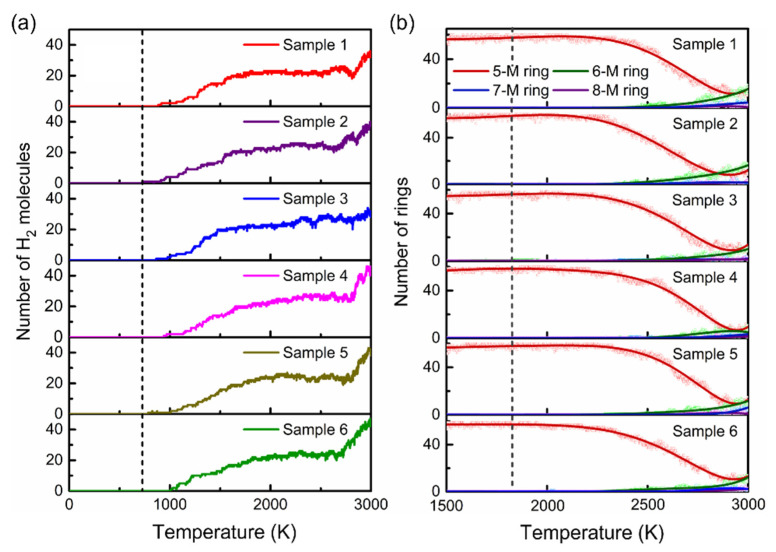
(**a**) Changes in the number of H_2_ molecules with the temperature during heating; (**b**) the number of rings in the six sets of samples versus temperature during heating.

**Figure 2 materials-15-03705-f002:**
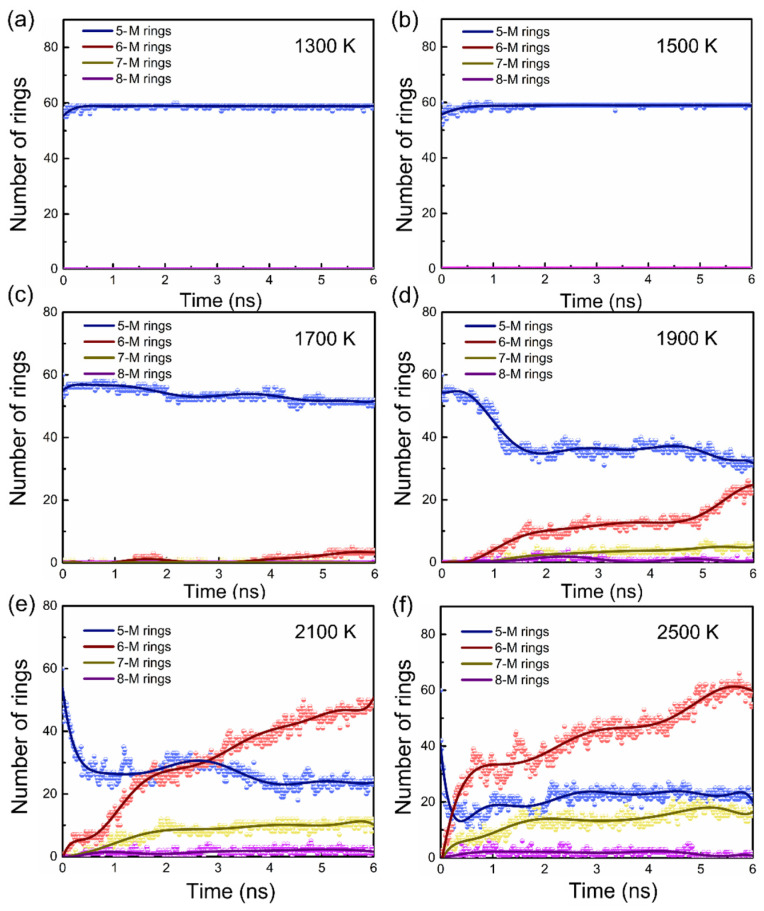
(**a**–**f**) are the variations of average ring numbers with temperatures from 1300 K to 2500 K. Below 1500 K, the type and number of rings in the system did not change. At 1700 K and above, the number of 5-M rings decreases, accompanied by the formation of new types of rings. As the temperature increases, the variety and number of new rings increased. The blue, red, yellow, and purple data sets represent the 5-M rings, 6-M rings, 7-M rings, and 8-M rings, respectively.

**Figure 3 materials-15-03705-f003:**
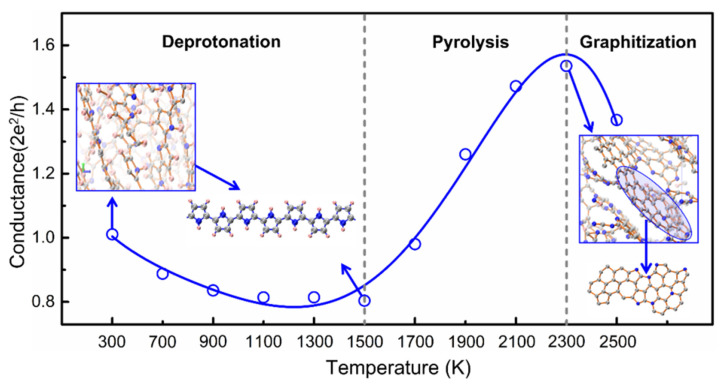
The variation of equilibrium statistical conductance with temperature. The tissue circled in blue is the graphite-like structure in the system. Carbon, nitrogen, and hydrogen atoms are represented by gray, blue, and pink spheres.

**Figure 4 materials-15-03705-f004:**
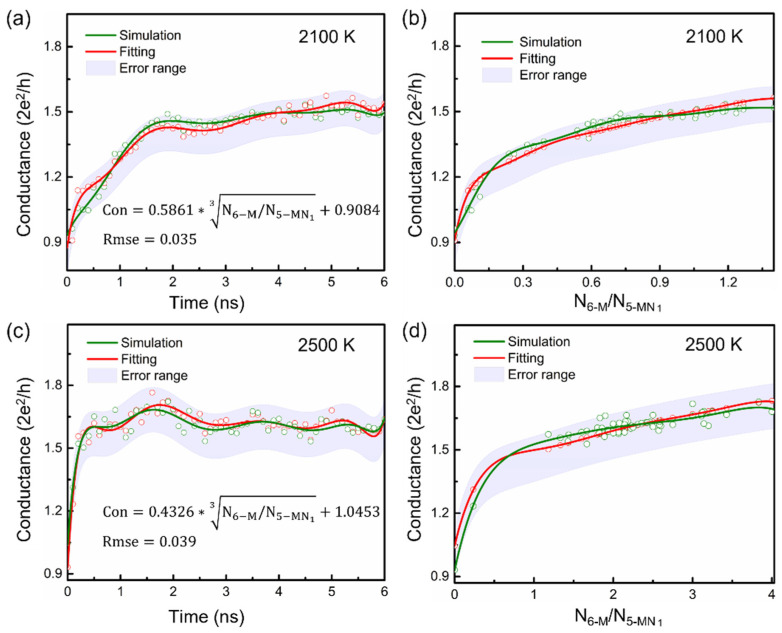
The statistical conductance variations with time and the structural factor at 2100 K (Subgraph (**a**,**b**)) and 2500 K (Subgraph (**c**,**d**)).

**Figure 5 materials-15-03705-f005:**
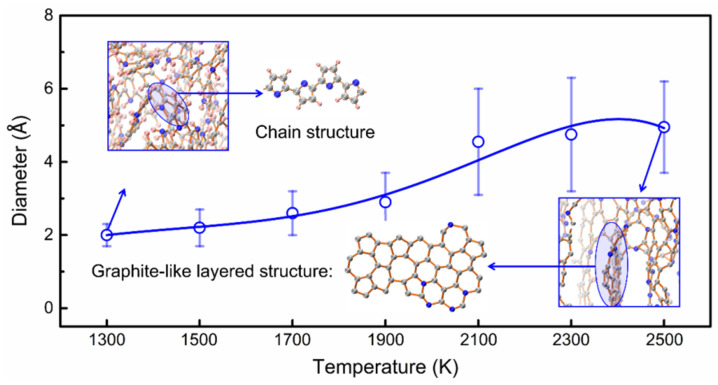
The variation of the equilibrium pore size distribution with temperature. The tissue circled in blue is the graphite-like structure in the system. Carbon, nitrogen, and hydrogen atoms are represented by gray, blue, and pink spheres, respectively.

**Figure 6 materials-15-03705-f006:**
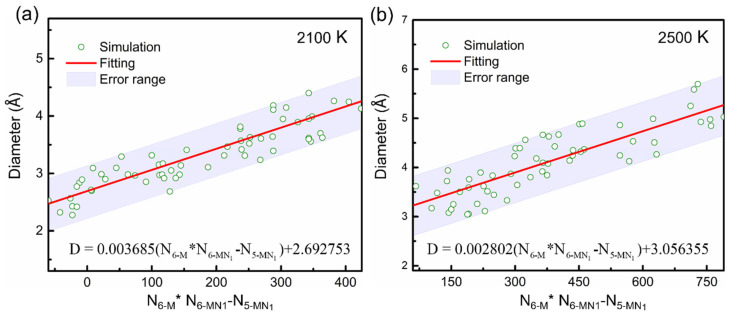
The pore size variations with the structural factor at 2100 K (**a**) and 2500 K (**b**).

**Figure 7 materials-15-03705-f007:**
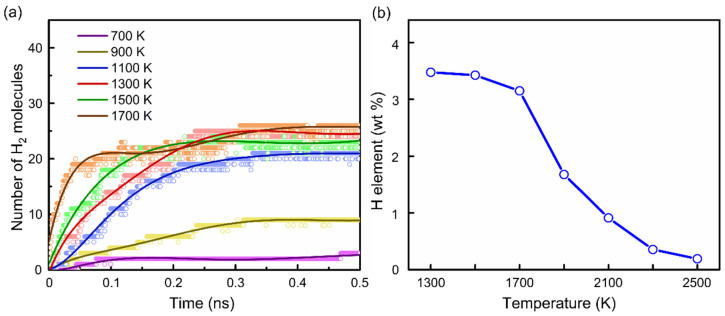
(**a**) Variation of the number of H_2_ molecules with time during the carbonization of PPy at 700–1500 K. (**b**) The change in the hydrogen content in PPy at different simulated temperatures.

**Figure 8 materials-15-03705-f008:**
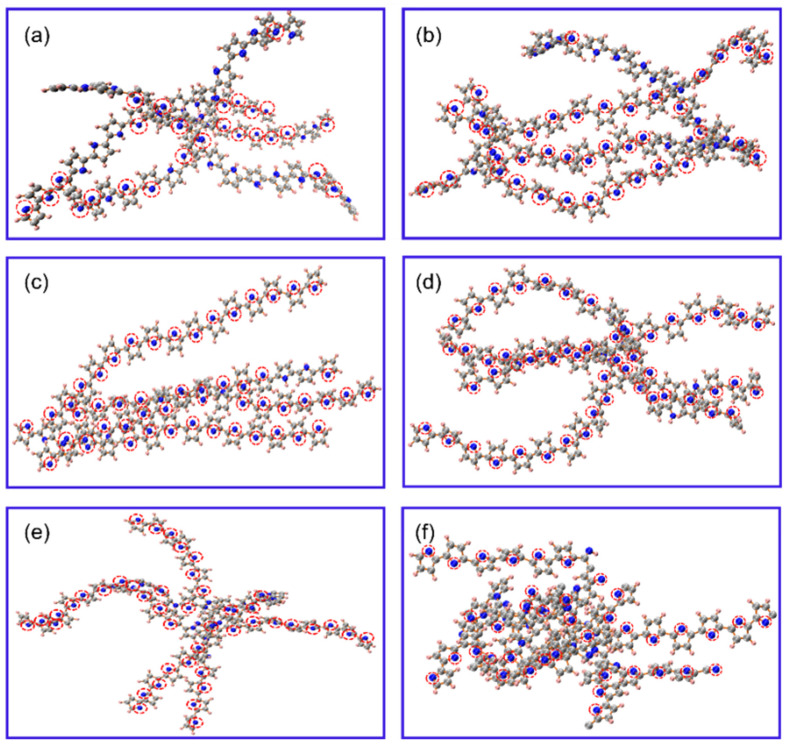
(**a**–**f**) are the respective equilibrium structures of carbonized PPy at the temperature from 700 K to 1700 K with an interval of 200 K. Red and hollow circles mark deprotonation sites. Carbon, nitrogen, and hydrogen atoms are represented by dark-gray, blue, and light-pink-colored spheres.

**Figure 9 materials-15-03705-f009:**
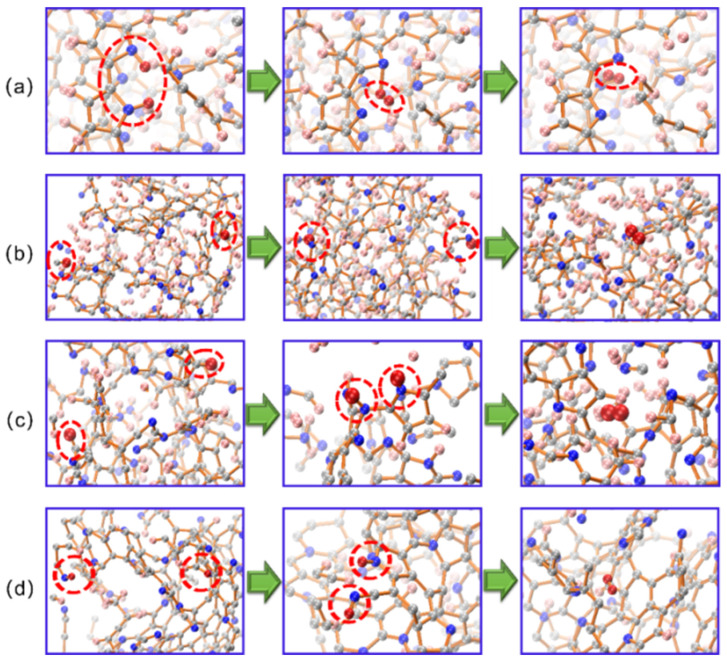
Hydrogen evolution paths observed in the MD trajectory at 1700–2500 K. (**a**–**d**) show that the hydrogen atoms with different initial positions first move to the nitrogen atoms and then generate H_2_.

**Figure 10 materials-15-03705-f010:**
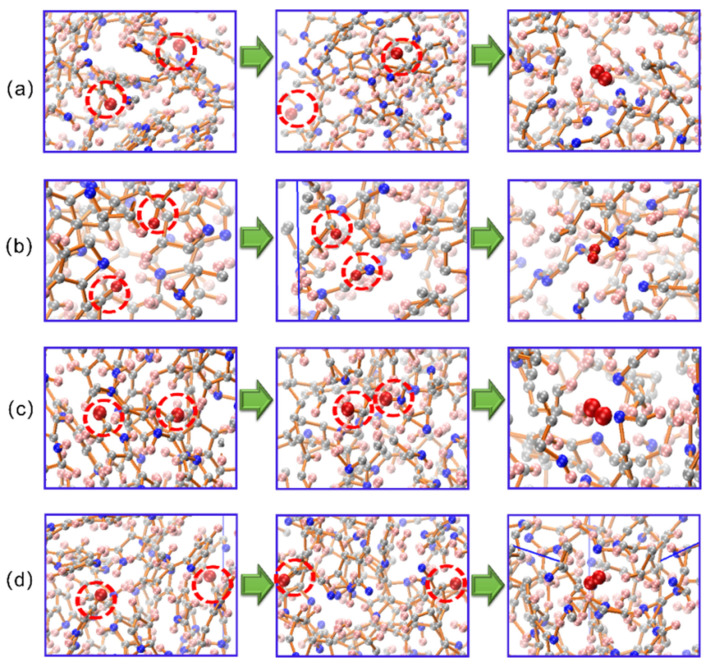
Hydrogen evolution paths observed in the MD trajectory at 2800 K. (**a**–**d**) indicate that the increase in temperature causes the direct dissociation of hydrogen at both the carbon and nitrogen sites.

**Table 1 materials-15-03705-t001:** The conductance (unit: 2e^2^/h) of different rings in the PPy carbonized tissue.

	5-M Ring	6-M Ring	7-M Ring	8-M Ring
N_0_	1.6319	2.0311	1.8404	1.1564
N_1_	0.9319	1.2667	2.1085	1.0176
N_2_	1.0053	1.4475	1.7509	0.9608
N_3_		1.8146		

N_0–3_: The number of nitrogen atoms in the carbon rings.

**Table 2 materials-15-03705-t002:** Weights of the effect of ring structures on conductance.

Temperature	Feature	* 5-M *	* 5-MN_1_ *	5-MN_2_	* 6-M *	* 6-MN_1_ *	6-MN_2_	6-MN_3_
**2100 K**	**Weights**	* 0.177 *	* 0.242 *	0	* 0.174 *	* 0.092 *	0.046	0.026
**2500 K**		* 0.139 *	* 0.211 *	0.007	* 0.201 *	* 0.177 *	0.047	0.003
**Temperature**	**Feature**	**7-M**	**7-MN_1_**	**7-MN_2_**	**8-M**	**8-MN_1_**	**8-MN_2_**	
**2100 K**	**Weights**	0.046	0.056	0.056	0.043	0.036	0.006	
**2500 K**		0.068	0.044	0.031	0.017	0.031	0.024	

**Table 3 materials-15-03705-t003:** Weights of the effect of ring structures on pore size.

Temperature	Feature	* 5-M *	* 5-MN_1_ *	5-MN_2_	* 6-M *	* 6-MN_1_ *	6-MN_2_	6-MN_3_
**2100 K**	**Weight**	* 0.187 *	* 0.224 *	0.018	* 0.140 *	* 0.132 *	0.052	0.032
**2500 K**		* 0.215 *	* 0.172 *	0.013	* 0.137 *	* 0.164 *	0.049	0.017
**Temperature**	**Feature**	**7-M**	**7-MN_1_**	**7-MN_2_**	**8-M**	**8-MN_1_**	**8-MN_2_**	
**2100 K**	**Weight**	0.052	0.029	0.052	0.026	0.033	0.023	
**2500 K**		0.051	0.066	0.035	0.028	0.032	0.021	

## Data Availability

Not applicable.

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
