# Peer review of "Microstructure Evolution and Its Correlation with Performance in Nitrogen-Containing Porous Carbon Prepared by Polypyrrole Carbonization: Insights from Hybrid Calculations"

_materials, 2022, doi:10.3390/ma15103705_

Round 1
Reviewer 1 Report
Review report on the topic ‘Microstructure Evolution and Its Correlation with Performance in Nitrogen-containing Porous Carbon Prepared by Polypyrrole Carbonization: Insights from Hybrid Calculations’. Comments are listed below:
- Strengthen the abstract section. Add the key conclusion of the works in the last two lines of the abstract section.
- Discuss the motive behind the work. The clear application of the work should be discussed in the introduction section.
- Reference style is also not uniform. Ex. ‘ Bairi et al. pointed out that the thermal conversion products are related to carbon 52 source and temperature [32-34].’
- There are numerous spelling and grammatical errors. Please revise the manuscript thoroughly. Sentences are also not complete and references are also cited in a rough manner.
- Try to make a bridge between current and previously published work and specify the gap area and objective of the work.
- Simulation work needs more clear discussion along with boundary conditions and equations.
- 1 describes the temperature and H2 molecule relations but the discussion is missing. Provide technical discussion behind the variation of the molecule with temperature.
- In present, it looks like a technical report. The results are presented without any technical discussion.
- Shorten the length of the conclusion section. Keep only key bullet points.
Reviewer 2 Report
To Authors,
Concerning the manuscript entitled “Microstructure Evolution and Its Correlation with Performance in Nitrogen-containing Porous Carbon Prepared by Polypyrrole Carbonization:Insights from Hybrid Calculations” (ID: materials-1702764) by Shanshan LI, Fang Bian, Xinge Wu, Lele Sun, Hongwei Yang, Xiangying Meng, Gaowu Qin:
Q1: In Fig. 2, the legend should be according to the data (colors). For instance, the yellow data set is for 7-M rings?
Q2: In Fig. 3 and 5, it is very hard to distinguish between gray and pink spheres.
Q3. How relevant is the machine learning model compared to the experimental data? It seems that N-doping can be controlled up to a point (e.g., 2500 K). Above that, not much is known from an experimental point of view. The sum of experimental work described in lines 305-314 falls within the machine learning model?
Q4. In Fig. 7, the curves in a) represent the H2 molecules that leave the structure at the given temperatures. What is represented in b)?.
Q5. The conclusion regarding the de-hydrogenation in the presence of nitrogen is essential.
Reviewer 3 Report
I read this work with an interest. The subject is of practical interest.
However, I have major concerns that need to be addressed
(1) "Experimental density". There are a number of definitions of density. Everything is clear with the mass, but not with the volume. Carbons obtained from polypyrrole consist of various micron and submicron sized "globules"/wires/tubes etc (e.g. doi: 10.1039/D0MA00730G ). The apparent volume of a sample actually includes the space between the globules. If the volume is obtained with He measurements, micropore volume may be interpreted as "empty space". The question is, whether the simulation setup truly takes into account the density definition it should. Otherwise the porosity the authors obtain in the course of the reaction might be an artefact of the simulation setup
(2) The machine learning methodology and its results and significantce are unclear (ground truth, training set test set...). The problem is, I did not find the supporting information (Table S1 of SI was mentioned). Please inform me where it is.
(3) the simulation time (~ns) is orders of magnitude shorter than the duration of actual processes of carbonization. I find the explanation of the projection of simulation results onto the real situation insufficient. How much do the sort-time simulations tell us on the real process?
(4) comparison with experiments: is conductivity the only quantity that can be compared? Because the structural comparison is down to purely qualitative level: such and such rings do exist
minor
sentence on line 26-28 is completely unclear "dihydrogen atom desorption"? Even without an "atom" this sentence is suitable to a catalytic process rather then the carbon structure formatio
Round 2
Reviewer 1 Report
Authors addressed my comments in the revised version. Now, manuscript can be accepted for publication.